# The Role of Ferroptosis in Adverse Left Ventricular Remodeling Following Acute Myocardial Infarction

**DOI:** 10.3390/cells11091399

**Published:** 2022-04-20

**Authors:** Kyoko Komai, Nicholas K. Kawasaki, Jason K. Higa, Takashi Matsui

**Affiliations:** 1Department of Anatomy, Biochemistry & Physiology, John A. Burns School of Medicine, University of Hawaii at Manoa, Honolulu, HI 96813, USA; komai3@hawaii.edu (K.K.); nkawasa2@hawaii.edu (N.K.K.); jkenjih@hawaii.edu (J.K.H.); 2Department of Microbiology and Immunology, School of Medicine, Keio University, Tokyo 160-8582, Japan

**Keywords:** ferroptosis, heart, iron, myocardial infarction, LV remodeling, heart failure, inflammation

## Abstract

Ferroptosis is an iron-dependent form of regulated cell death and is distinct from other conventional forms of regulated cell death. It is often characterized by the dysfunction of the antioxidant selenoprotein glutathione peroxidase 4 (GPX4) antioxidant system. This loss of antioxidant capacity leads to the peroxidation of lipids and subsequent compromised plasma membrane structure. Disruption of the GPX4 antioxidant system has been associated with various conditions such as cardiomyopathy and ischemia-reperfusion (I/R) injury. GPX4 regulates lipid peroxidation, and chemical or genetic inhibition of GPX4 leads to reduced cardiac function. Iron chelators or antioxidants can be used for inhibiting ferroptosis, which restores functionality in in vivo and ex vivo experiments and confers overall cardioprotective effects against I/R injury. Moreover, suppression of ferroptosis also suppresses inflammation and limits the extent of left ventricle remodeling after I/R injury. Future research is necessary to understand the role of ferroptosis following an ischemic incident and can lead to the discovery of more potential therapeutics that prevent ferroptosis in the heart.

## 1. Introduction

Ferroptosis is a new form of regulated cell death characterized by the inactivation of glutathione peroxidase 4 (GPX4) and the accumulation of lipid peroxides that disrupt cell membrane integrity [1]. These pathophysiological changes are not found in other forms of regulated cell death, including apoptosis, necroptosis, and pyroptosis [2]. Regulated cell death has been well studied in cardiac diseases, including acute myocardial infarction (MI) and heart failure [3]. However, the role of ferroptosis in the heart is still yet to be well characterized due to a limited number of studies that specifically use models of cardiovascular disease.

Compared with more well-known cell death mechanisms, such as apoptosis, necrosis, and necroptosis, ferroptosis is a relatively newly discovered form of regulated cell death. The initial studies that eventually led to the discovery of ferroptosis did not start with investigating specific cellular pathways, but rather with the screening of compounds that cause cell death. Erastin and RAS-selective lethal compound 3 (RSL3) were first obtained from the screening of compounds that caused tumor cell death, and it was only later that they were both found to specifically cause ferroptosis [4,5,6]. Dixon et al. published the first study naming ferroptosis and providing the first details of its mechanism. In that study, they showed that erastin has inhibitory effects on system Xc^−^, the cystine/glutamate antiporter, and that erastin-induced cell death has characteristics unlike other known forms of cell death. Dixon et al. went on to call this new form of cell death ferroptosis, based on the unique iron-dependent accumulation of cytosolic and lipid ROS that distinguishes ferroptosis contrasts it with other types of cell death [1]. The implication of system Xc^−^ in ferroptosis led to more studies investigating the antiporter and how altering its activity causes downstream events that lead to ferroptosis and cell death. A study in neuronal cells demonstrated that cystine uptake through system Xc^−^ plays a vital role in cell survival against oxidative stress and does so by regulating cellular levels of reduced glutathione (GSH) [7,8]. GPX4, a selenoprotein, uses glutathione as a substrate and prevents lipid peroxidation by reducing peroxides and oxidizing GSH into glutathione disulfide (GSSG) in the process [9]. Further studies have revealed that glutathione peroxidase 4 (GPX4) is a target of RSL3 [10], which led to the elucidation of the GPX4 anti-lipid peroxidation pathway as the central regulatory mechanism of ferroptosis.

Research on pathways that promote lipid peroxidation has also progressed and provided further insight into the downstream effects of GPX4 inhibition and ferroptosis. Polyunsaturated fatty acids (PUFAs) are oxidized during ferroptosis, and the oxidation of phospholipids containing PUFAs leads to a change in cell membrane structure. Increased PUFA oxidation leads to increased curvature of membranes, which eventually leads to destabilization and pore formation in the cell membrane [11]. To this effect, it has been shown that adding polyethylene glycols larger than the membrane pores formed during ferroptosis was sufficient to reduce cell death induced by ferroptosis [12]. The modulation of enzymes involved in the metabolism of PUFAs and phospholipids has also been shown to affect ferroptosis. Acyl-CoA synthase long-chain family member 4 (ACSL4) is an enzyme required for the incorporation of arachidonic acid—a PUFA—into phospholipids [13,14], and 15-lipoxygenase (15-LOX) [15], which promotes the production of phospholipid hydroperoxide (PLOOH), have both been reported as factors that promote phospholipid peroxidation and thus ferroptosis. GPX4 uses GSH as a substrate and oxidizes GSH into GSSG to reduce phospholipid hydroperoxide (PLOOH) radicals into phospholipid alcohol (PLOH), thereby stopping the chain reaction of lipid peroxidation [16].

Since its discovery, ferroptosis has been reported in multiple cell types, tissues, and organs, including the kidney [17,18], liver [15], neural cells [19], and cardiomyocytes [20,21,22,23]. Ferroptosis has also been reported to be associated with many diseases, especially those involving ischemia [24]. The involvement of ferroptosis in both cardiomyocytes and ischemia makes it an attractive potential target in preventing cell death in cardiac tissues during an ischemic event, such as a myocardial infarction (MI). A myocardial infarction occurs due to a blockage in a coronary artery, resulting in ischemia and the loss of cardiac cells. The loss of functional cells results in overall cardiac pump dysfunction along with subsequent pathological adaptations occurring in response to cardiac cell death and loss of cardiac function. One such adaptation is adverse left ventricular (LV) remodeling, which occurs following an acute MI and is characterized by LV dilatation and fibrosis. Adverse LV remodeling is a critical determinant of the subsequent development of heart failure, which significantly affects life expectancy and quality of life [25,26]. The magnitude of cardiomyocyte cell death caused by ischemic injury is directly proportional to the future risk of LV remodeling [27]. Therefore, one strategy to prevent adverse LV remodeling and consequent heart failure is to go upstream and prevent or limit cell death, which is the initial incident that sets off that chain of pathological events. Thus, developing and employing therapies that prevent cardiomyocyte cell death decreases LV remodeling and prevents the development of heart failure. In addition to the initial cell death caused by myocardial injury, inflammation also contributes to the pathogenesis of post-MI LV remodeling. Ferroptosis also plays a role in post-MI inflammation [28]; therefore, inhibition of ferroptosis may prevent both cell death and inflammation in the heart, yielding a potential double-pronged approach to prevent post-MI LV remodeling and subsequent heart failure.

Reperfusion therapies, such as primary percutaneous coronary intervention (PCI), can reduce the initial infarct size in patients presenting with an acute MI. However, the reperfusion therapy itself induces further injury in the myocardium [29]. Therefore, understanding the mechanisms of ischemia-reperfusion (I/R) injury, and not just ischemia alone, is essential for attenuating cardiomyocyte cell death and preventing LV remodeling and heart failure. Recent reports suggest that ferroptosis is involved in the initial myocardial cell death caused by I/R injury (Figure 1) [23,30]. In this review, we will discuss the pathophysiological role of ferroptosis in adverse LV remodeling following I/R injury.

## 2. Features and Mechanisms

Ferroptosis is a type of regulated cell death that presents with morphologically unique cellular features such as mitochondrial shrinkage and increased mitochondrial membrane density [1]. There are two primary pathophysiological mechanisms that cause these changes: (1) iron-dependent lipid peroxidation and (2) loss of antioxidant system homeostasis. These result in cell death due to oxidative imbalance and damage to lipids constituting the cell membrane (Figure 1). These mechanisms are also unique to ferroptosis, and neither of them is a central mechanism in apoptosis, necroptosis, or other forms of cell death. The major cellular changes that differentiate each type of cell death have been previously discussed at length [31].

### 2.1. Lipid Peroxidation and Iron Homeostasis

The phospholipids found in cell membranes are composed mainly of phosphate and varying types of fatty acids, including saturated and unsaturated fatty acids. Since unsaturated fatty acids are easily oxidized, phospholipids containing polyunsaturated fatty acids (PUFAs) are often involved in lipid oxidation pathways. The activity of enzymes that regulate PUFA metabolism and oxidation can also trigger ferroptosis. For example, iron-dependent lipoxygenase activity that results in the oxidation of PUFAs is necessary for ferroptosis [32]. This is supported by studies investigating the effects of deleting acyl-CoA synthase long-chain family member 4 (ACSL4), an enzyme required for metabolic pathways that incorporate arachidonic acid or adrenoyl acid—a PUFA—into phospholipids. These studies found that ACSL4 deletion provides protection against ferroptosis [13,14]. Activation of AMP-activated protein kinase (AMPK) has been shown to reduce ferroptosis through inhibition of acetyl-CoA carboxylase and subsequent reduction in PUFAs [33]. Lipidomic analysis showed that the absence of AMPK increased PUFA, suggesting that AMPK is involved in PUFA biosynthesis and regulation of ferroptosis.

Iron is an essential mineral for maintaining biological functions in physiological conditions. However, iron can also cause potentially toxic effects in multiple organs, including the heart [34,35]. In ferroptosis, intracellular hydroxyl radicals (•OH) are overproduced as a result of the Fenton and Fenton-like reactions between H_2_O_2_ and free ferrous labile iron (Fe^2+^). The accumulation of free radicals triggers a chain of events that lead to lipid peroxidation [36]. Iron is an essential component in this process, as sequestration of iron with chelators has been shown to inhibit erastin-induced ferroptosis, suggesting that iron is required for ferroptosis [1].

The intracellular iron that catalyzes ferroptotic mechanisms is supplied by two major processes that affect intracellular iron homeostasis (Figure 2): (1) the uptake of circulating iron into the cell and (2) the degradation of the intracellular iron-storing protein, ferritin. These processes add or release more Fe^2+^ into the labile iron pool (LIP) of the cytoplasm [37].

The first process involves cellular uptake of iron and requires the binding of transferrin to circulating ferric iron (Fe^3+^). The transferrin-bound ferric iron complex can then bind to transferrin receptor 1 (TfR1), which mediates cellular uptake of transferrin and iron into endosomes, from which iron is released into the cytoplasm and LIP [38]. This iron uptake mechanism has a potential role in ferroptosis, as in the inhibition of TfR1 with short hairpin RNA (shRNA) suppressed ferroptosis [21]. Divalent Metal Transporter 1 (DMT1) is another transmembrane protein responsible for the uptake of non-transferrin-bound ferrous iron (Fe^2+^) into cardiomyocytes [39,40]. DMT1 also transports Fe^2+^ from endosomes into the cytoplasm and LIP, and increased expression of DMT1 has been associated with an increase in the amount of Fe^2+^ available for redox reactions [41,42]. Subsequently, inhibition of DMT1 by siRNA leads to decreased iron in the LIP and decreased ROS production [43].

The second process involves the degradation of ferritin, a protein that sequesters and stores iron in the cell. Ferritinophagy—the autophagic degradation of ferritin—increases free iron in LIP, thus increasing cellular sensitivity to ferroptosis [44]. Inhibition of ferritinophagy has been reported to inhibit the release of iron into the cytoplasm and also inhibit ferroptotic cell death [45]. In cancer cells, inhibition of bromodomain-containing protein 4 (BRD4), a transcriptional regulator that promotes tumor growth, induces ferroptosis by increasing ferritinophagy [46]. Thus, ferroptosis can be triggered by the accumulation of Fe^2+^ in the cytoplasm via changes in the activity of processes that regulate iron uptake and the release of intracellular iron stores.

### 2.2. Disruption of Antioxidant Systems

The antioxidant system is a network of metabolic pathways that maintain overall redox homeostasis in the cell and inhibit the ROS-initiated intracellular chain reactions that lead to lipid peroxidation.

#### 2.2.1. GPX4-Dependent Antioxidant Mechanisms

GPX4 is a selenoprotein that belongs to the family of glutathione peroxidases, which protect cells from oxidative damage [47]. The presence of selenocysteine in the active site of GPX4 is necessary for the efficient removal of peroxides, as evidenced by comparing the effects of cysteine (Cys) versus selenocysteine (Sec) utilization in GPX4 [48]. System Xc^−^ is a cystine/glutamate antiporter responsible for the uptake of cystine, which is required for the synthesis of glutathione (GSH) molecules needed as a substrate for GPX4-mediated antioxidant mechanisms (Figure 2). Before cystine can be integrated into its final functioning product, GSH, cystine must be reduced into cysteine by thioredoxin reductase 1, which is an integral step in GSH synthesis [49]. Cysteine, along with glycine and glutamate, participates in GSH synthesis. Therefore, inhibition of system Xc^−^ will decrease cystine uptake and deprive cells of one of the amino acids needed for GSH synthesis, resulting in a lack of substrate and GPX4 activity [10]. Erastin triggers ferroptosis by binding to and inhibiting system Xc^−^ and VDAC 2/3, thus causing cell death through lipid peroxidation resulting from a lack of GSH synthesis and GPX4 activity [5]. Inhibition of glutaminase (GLS), which converts L-glutamine to glutamate, also causes ferroptosis through inhibition of GSH synthesis, which requires both glutamate and cysteine [21].

GPX enzymes use GSH to donate electrons and reduce substrates [50]. Therefore, an adequate supply of GSH is required to maintain the antioxidant activity of GPX [47]. RSL3, a class 2 ferroptosis inducer, directly inhibits GPX4, thereby increasing lipid peroxidation and inducing ferroptosis [32]. A study using inducible GPX4 knockout mice showed that inactivation of GPX4 resulted in ferroptotic cell death, and that the GSH-GPX4 axis is necessary to prevent lipid oxidation-induced acute renal failure [18]. Mice born with a mutated selenocysteine synthase (SEPSECS), an enzyme required for forming selenocysteine residues, were shown to have lethal cardiopulmonary dysfunction. Mice with deficient SEPSECS activity were rescued through independent gene expression of GPX4, showcasing the critical role of the antioxidant activity of GPX4 in the heart [51].

#### 2.2.2. GPX4-Independent Antioxidant Mechanisms

Although research investigating the role of antioxidant systems in ferroptosis has mostly focused on GPX4, other novel cellular mechanisms have started to come into the picture. For example, ferroptosis suppressor protein 1 (FSP1), previously known as apoptosis-inducing factor mitochondria-associated 2 (AIFM2), has recently been reported as a new antioxidant by two groups [52,53]. FSP1 is recruited to the plasma membrane by N-terminal myristoylation and reduces ubiquinone (CoQ_10_) to ubiquinol (CoQ_10_H_2_) (Figure 2). Ubiquinol is a lipophilic radical-trapping antioxidant that prevents the accumulation of lipid ROS [52,53]. Thus, it has been shown that disruption of the reductive activity of FSP1 and ubiquinol triggers a chain reaction of excessive lipid peroxidation, leading to ferroptosis.

#### 2.2.3. Mitochondria-Dependent Ferroptosis

Mitochondria are involved in various types of cell death [54]. Initial studies of ferroptosis demonstrated that mitochondria were not necessary for the induction of ferroptosis [55]. However, recent studies revealed the existence of mitochondria-dependent mechanisms of ferroptosis [56,57].

One such study reported that in cells lacking mitochondria, decreased lipid ROS formation and less cell death were observed in response to erastin as compared to cells with functional mitochondria subjected to the same treatment. The study went on to show that cysteine deprivation induces ferroptosis through the accumulation of ROS produced by the mitochondrial electron transport system and that inhibition of the electron transport system suppresses cell death [58]. Another report showed that oxidized phosphatidylethanolamines (OxPEs) cause ferroptosis [14] and that OxPEs accumulate in the mitochondria and cytosol in response to RSL3 treatment. In that same report, mitochondria-targeted ROS scavengers were able to reduce the increased OxPE formation seen in ex vivo cardiac I/R injury. The study also showed that inhibition of GSH transport to the mitochondria caused an increase in mitochondrial ROS and increased cell death [59].

In doxorubicin (DOX)-induced cardiomyopathy, downregulation of GPX4 and an increase in lipid peroxidation were associated with the formation of DOX-Fe^2+^ complexes observed in mitochondria [60]. This doxorubicin-induced cell death was inhibited by a mitochondria-specific Fe^2+^ chelator, further implicating ferroptosis. It was reported that doxorubicin-induced downregulation of GPX4 and cell death are suppressed by MITOL, an E3 ubiquitin ligase that plays an important role in the regulation of mitochondrial quality and function [61].

Another notable mitochondrial mechanism is detailed in a report investigating the anti-ferroptotic properties of the mitochondrial enzyme dihydroorotate dehydrogenase (DHODH). DHODH is an enzyme localized on the intermembrane space surface of the mitochondrial inner membrane, and it reduces lipid peroxide formation and protects against ferroptosis by reducing CoQ_10_ to CoQ_10_H_2_. In one of the experiments in this study, the investigators tried to rescue DHODH knockout cells by transfecting and reintroducing DHODH, but instead used a DHODH mutant that could not localize to the mitochondria. The mutant DHODH failed to rescue those cells and prevent cell death, further emphasizing the role and location of the mitochondria in the typically anti-ferroptotic activity of DHODH. Furthermore, inactivation of both DHODH and GPX4 triggered ferroptosis by inducing mitochondrial lipid peroxidation much more potently than GPX4 alone [62].

Although ferroptosis can be induced in cells without the direct involvement of mitochondria, several groups have identified mitochondria-dependent mechanisms of ferroptosis. In light of this, further research specifically targeting mitochondrial pathways and metabolism is expected.

## 3. Ferroptosis in I/R Injury and LV Remodeling

### 3.1. Ferroptosis in Cardiomyocyte Cell Death

#### 3.1.1. Role of Iron in I/R Injury

Myocardial injury can lead to the release and accumulation of iron in the affected tissue [63]. T2 Star (T2*) on cardiac magnetic resonance imaging (CMR) can noninvasively predict myocardial iron levels in myocardial hemorrhage in patients with acute MI [64]. The presence of myocardial hemorrhage is a useful prognostic indicator of adverse LV remodeling in patients with an MI reperfused with PCI [65]. The reactive oxygen species generated by this accumulated iron in cardiac tissue can trigger pathological events such as cell death and inflammation. Thus, it is likely that ferroptosis occurs during cardiac I/R injury and that ferroptosis is involved in the exacerbation of the disease.

Iron chelators have been used clinically to treat iron-loading disorders and are expected to be a therapeutic strategy for the prevention of ferroptosis [66]. Deferoxamine (DFO) is a clinically approved iron chelator with a very high affinity for binding Fe^3+^. Administration of DFO has shown cardioprotective effects by suppressing cytosolic ROS production in ex vivo I/R models using mouse hearts [21]. Dexrazoxane (DXZ), which is also a chelator of Fe^3^^+^, showed cardioprotective effects by inhibiting ferroptosis in in vivo I/R injury models in mice [23].

As discussed above, transferrin receptor 1 (TfR1) is one of the key factors that regulate iron homeostasis in the cell. A recent study showed that upregulation of ubiquitin-specific protease 7 (USP7) caused alterations in p53 and TfR1 activity that were concomitant with an increase of ferroptosis in rat I/R models [67]. This study also used pharmacological agents to show that inhibition of USP7 suppressed deubiquitination, which in turn increased p53 activity and downregulated TfR1, thus resulting in decreased ferroptosis and decreased myocardial I/R injury. Therefore, the use of pharmacological drugs that downregulate TfR1 or target upstream factors that decrease TfR1 activity may provide protection against ferroptosis.

Nuclear receptor coactivator 4 (NCOA4) promotes the release of Fe^2+^ into the cytoplasm by binding to ferritin and forming autophagosomes used in ferritinophagy [44]. In cardiomyocyte-specific NCOA4 deficient mice, ferroptosis was suppressed in pressure-overload-induced cardiomyopathy by inhibiting ferritinophagy, which preserved the cardiac function of the mice [44]. The role of autophagy in ferroptosis is also reported in other signaling pathways. In the context of liver fibrosis, the RNA-binding protein embryonic lethal-abnormal vision-like protein 1 (ELAVL1) plays a crucial role in regulating ferroptosis via an increase in autophagy [68]. The Forkhead box C1 (FOXC1) protein is a transcription factor that belongs to the FOX family and regulates ELAVL1. FOXC1 also promotes ferroptosis in I/R injury via increasing autophagy, thus underscoring the potential role of ELAVL1 in ferroptosis [69]. Taken together, inhibition of factors that promote ferritinophagy like NCOA4 and ELAVL1 may be another avenue for preventing ferroptosis in the heart.

Ferroportin is a transmembrane protein that transports iron from intracellular to extracellular sites [70]. Depletion of ferroportin in mouse cardiomyocytes has been reported to increase iron accumulation in the heart and cause heart failure [71]. Conversely, overexpression of ferroportin has been shown to reduce ROS in breast cancer [72]. A study using BTB domain and CNC homolog 1 (BACH1) knockout mice demonstrated that increasing iron ferroportin led to a decrease in intracellular labile iron concentrations, thus protecting the heart against I/R injury by inhibiting ferroptosis [73]. Since ferroportin affects intracellular iron homeostasis, finding ways to increase ferroportin expression or transport may be a potential therapeutic target for ferroptosis-related diseases in the heart.

Nuclear factor erythroid 2-related factor 2 (Nrf2) is a redox-sensitive transcription factor that regulates the expression of genes in normal and stressed conditions, including several proteins that regulate ferroptosis, such as heme oxygenase 1 (HMOX1) [74]. A previous study using Nrf2 knockout mice showed that in doxorubicin-induced cardiomyopathy models, the Nrf2/HMOX1 pathway plays an important role in cardiac injury caused by ferroptosis [23]. A recent paper also suggested that in rat I/R injury models, the Nrf2 pathway contributes to ferroptosis in I/R injury [75]. The well-established role of the Nrf2 pathway in antioxidative defenses makes it an attractive target for cardiomyocyte protection against the oxidative stresses seen during ferroptosis and I/R injury.

In addition to ferroptosis, apoptosis and necroptosis are also observed in I/R injury. However, there are limited studies implicating when and how much each type of cell death contributes to I/R injury. A study using rat in vivo I/R models demonstrated that ferroptosis occurs mainly in the phase of myocardial reperfusion, but not ischemia [30]. The investigators also detected ferroptosis by observing an increase in the levels of ACSL4, iron, and malondialdehyde (MDA), which accompanied a decrease in the level of GPX4 expression. Interestingly, while caspase 3 cleavage is a hallmark of apoptosis not seen in ferroptosis [1], TUNEL assays (which detect fragmented DNA histone units) stain cardiomyocytes that undergo either ferroptosis or apoptosis [60]. In light of this overlap, more specific markers and assays for ferroptosis need to be developed to better distinguish it from other forms of cell death and to characterize how much each type of cell death contributes to the overall I/R injury.

#### 3.1.2. Antioxidant System Disruption in I/R Injury

Overexpression of GPX4 in mitochondria suppresses I/R injury and reduces lipid peroxidation [76]. Inactivation of GPX4 using siRNA has been shown to induce lipid peroxidation and ferroptotic cell death in cardiomyocyte cell lines [77]. Proteomic analysis in mice models of MI revealed that GPX4 is markedly decreased during MI [77]. It has been reported that in rodent models, decreases in GPX4 occur during reperfusion rather than during ischemia [67,78]. The importance of GPX4 activity and GSH in antioxidant homeostasis combined with the evidence showing dysregulation of GPX4 during I/R injury suggest that GPX4 has a powerful role in preventing oxidative stress in I/R injury. Thus, manipulating GPX4 expression and increasing its activity in the heart is an attractive and potentially powerful therapeutic target for potential new therapies that regulate ferroptosis during I/R injury.

Solute carrier family 7 member 11 (SLC7A11) is a subunit of the structure constituting the system Xc^−^ transporter and is transcriptionally repressed by the tumor suppressor p53. Overexpression of SLC7A11 is known to suppress ROS-induced ferroptosis [79]. A recent report showed that in mice lacking ferritin H (Fth) in the myocardium, SLC7A11 expression was suppressed, resulting in high-iron diet-induced cardiac hypertrophy and dysfunction. However, these Fth-deficient mice were rescued by treatment with Fer-1, a ferroptosis inhibitor. Furthermore, SLC7A11 overexpression in cardiomyocytes increased GSH levels and rescued the cardiac dysfunction in Fth conditional knockout mice. Targeting SLC7A11 and increasing its activity represents a new therapeutic possibility that works upstream of GPX4 and GSH activity [80].

#### 3.1.3. Lipid Peroxidation in Ischemia-Reperfusion

As mentioned above, excess Fe^2+^ can generate lipid radicals independently of enzymatic activity and trigger a chain reaction of lipid peroxidation. The formation of lipid peroxides such as PLOOH is a key factor in initiating the chain reactions of lipid peroxidation. PLOOHs are reduced by GPX4, which serves to halt the chain reactions that lead to lipid peroxidation. However, PLOOHs can also be generated by reactions catalyzed by enzymes such as 15-lipoxygenase (15-LOX). Inhibition of 15-LOX has also been reported to inhibit ferroptosis [15,81], suggesting that lipoxygenase inhibitors may also be a promising class of drugs to investigate in inhibiting ferroptosis in I/R injury and other cardiovascular diseases.

In addition to chelators and enzymatic inhibitors of lipid peroxide formation, natural antioxidants such as vitamin E and synthetic antioxidant compounds have been found to inhibit lipid peroxidation and ferroptosis. Ferrostatin (Fer-1) is a synthetic antioxidant and specific inhibitor of ferroptosis that inhibits the formation of lipid peroxides [82]. Administration of Fer-1 reduced the size of acute infarct lesions caused by I/R injury in mice and showed long-term improvement in cardiac function [22]. Liproxstatin-1 (Lip-1), another anti-ferroptotic antioxidant, reduced mitochondrial ROS production and rescued the cardiac I/R stress-induced decrease in antioxidant GPX4 in mice [78].

Nicotinamide adenine dinucleotide phosphate (NADPH) oxidase (NOX) is the main generator of ROS during myocardial I/R injury. In models of in vivo I/R and in vitro hypoxia/reperfusion injury, levels of NOX2 increased and were further accentuated in diabetic rat models [83]. Inhibition of NOX2 in the I/R rat model resulted in an increase in GPX4, suggesting that NOX2 has a role in ferroptosis. NOX2 was also elevated in the absence of AMPK, and AMPK depletion attenuated GPX4 expression, suggesting that AMPK also suppresses ferroptosis in the heart [83].

OxPEs are phospholipids that have been suggested as a cause of ferroptotic cell death [14]; however, few studies have directly analyzed OxPE formation during ferroptosis or in response to cardiac I/R injury or treatment with anti-ferroptotic compounds. However, the few studies available have shown increased OxPE formation in ex vivo cardiac I/R injury [59], and that Fer-1 reduces OxPE in heart grafts [22]. Another study used gas cluster ion beam secondary ion mass spectrometry (GCIB-SIMS) imaging to analyze OxPEs instead of liquid chromatography-mass spectrometry (LC-MS) and visualized increased OxPE formation in myocardial cell lines treated with RSL3 [84].

Oxidized phosphatidylcholines (OxPCs) are phospholipids that are elevated in rat I/R injury models [85]. A recent study demonstrated that the administration of fragmented OxPCs caused cell death in in vitro rat cardiomyocytes, which was accompanied by impaired Ca^2+^ transients, reduced GPX4 activity, and disrupted mitochondrial respiration [86]. The study showed that treatment with Fer-1 or Lip-1 prevented cell death caused by fragmented OxPCs, suggesting that fragmented OxPCs induce ferroptosis [86]. Taken together, inhibiting the formation or effects of pro-ferroptotic phospholipids like OxPEs and OxPCs may be a potential strategy for new therapeutics that protect against I/R injury by inhibiting ferroptosis.

#### 3.1.4. Identification of Ferroptosis in Cardiomyocytes

While no single specific downstream marker for ferroptosis has been identified using current assays and techniques, ferroptosis is instead detected by assessing multiple overall changes in oxidative stress and lipid peroxidation, and these methods have been used to identify ferroptosis in cardiomyocytes [87]. As discussed above, levels of GSH and GPX4 activity are unique features of ferroptosis compared to other forms of cell death. The ratio of the reduced and oxidized forms of glutathione (GSH/GSSG) regulated by GPX4 is considered a marker of oxidative stress and ferroptosis. The expression level of GPX4 also reflects its overall enzymatic activity [10].

Prostaglandin-endoperoxide synthase-2 (PTGS2) mRNA expression is upregulated in a human skin-derived cell line upon treatment with either erastin or RSL3 [10]. PTGS2 upregulation may be a suitable marker for lipid peroxidation in ferroptosis induced by GPX4 suppression [10]. This potential marker has also been seen in the heart, as an increase in PTGS2 was observed in doxorubicin-induced cardiomyopathy [23] and ischemia-reperfusion injury [23].

Liperfluo [14] and C11-BODIPY [88] are lipophilic fluorescent probes used to detect increases in lipid peroxide formation in histological assays. The probes associate with lipid bilayers and fluoresce when they are oxidized by lipid peroxides. As described above, 4-hydroxynonenal (HNE) and malondialdehyde (MDA) are specific lipid peroxides observed during ferroptosis and are used as another type of marker to detect ferroptosis [87].

Disruption in iron homeostasis is another way to identify ferroptosis, and both iron transporters and free iron levels may be measured. An increase in the expression level of transferrin receptor 1 (TfR1) can be observed in I/R injury [67]. Increased intracellular iron content also causes ferroptosis and can either be measured directly through cell lysates or as detected by decreased GPX4 activity and increased ACSL4 expression [67].

### 3.2. Ferroptosis and Inflammation during I/R Injury

The LV remodeling that occurs after an ischemia-reperfusion injury is a major predictor of long-term prognosis [26]. Inflammatory cells play a significant role in the remodeling process [28], and are triggered by inflammatory molecules released from stressed or dead cells. These pro-inflammatory molecules serve as “danger” signals to inflammatory cells, which coordinate and activate other cells involved in the repair and the remediation of injured tissues [89]. Damage-associated molecular patterns (DAMPs) are a category of pro-inflammatory molecules that are released from dead or dying cells and include intracellular proteins and nucleic acids. DAMPs are recognized by innate immune cells through pattern recognition receptors (PRRs) such as Toll-like receptor 4 (TLR4), which is a potent initiator of inflammatory pathways [90].

#### 3.2.1. Inflammation Following Ferroptosis

DAMPs have also been reported to be released from ferroptotic cells. Ferroptosis caused by renal I/R injury has been shown to release mitochondria and nuclei [18], and HMGB1 has been reported to be released from cancer cell lines undergoing ferroptosis [91]. Since cytoplasmic components contain a variety of DAMPs, current knowledge posits that if the cell membrane is damaged by ferroptosis, intracellular DAMPs will be released due to the compromised cell structure, triggering an immune response [92]. In a model of renal injury, suppression of ferroptosis was reported to suppress leukocyte extravasation [17]. After cardiac transplantation and cardiac I/R injury, vascular endothelial cells have been reported to release type 1 IFN in response to Toll-like receptor adaptor molecule 1 (TICAM1) and TLR4 signaling. This activation of innate Toll-like receptor inflammatory responses caused neutrophil accumulation and myocardial damage and was reversed and ameliorated by Fer-1 administration [22]. In addition to this, 4-hydroxynoneal (4-HNE), a product of lipid peroxidation, has also been reported to be a ligand for TLR4 receptors and trigger inflammation mediated by TLR4 signaling [93].

Taken together, these studies suggest that there is a relationship between ferroptosis, DAMPs, and inflammation. This possible relationship is further supported by studies that show that inhibition of ferroptosis by Fer-1 improves LV remodeling after cardiac I/R failure, which suggests that inhibition of ferroptosis affects cardiac remodeling through the control of inflammation [22,23]. However, it is still unclear what DAMPs are actually released during ferroptosis in cardiac I/R injury, and a full mechanism of the progression of inflammation triggered by DAMPs in the context of I/R injury is yet to be published [94]. We expect further analysis of DAMPs released from ferroptotic cells and subsequent inflammation induced by cardiac I/R injury to be subjects of future studies that will help to elucidate a full mechanism linking ferroptosis and activation of innate inflammatory pathways in I/R injury.

#### 3.2.2. Ferroptosis and Inflammatory Cells

In addition to studies demonstrating how ferroptosis initiates and activates inflammatory pathways, other studies have shown that inflammatory cells can also regulate ferroptosis, especially in the context of cancer. It has also been reported in hepatocellular carcinoma cell lines that the cytokine TGF-beta suppresses SLC7A11 via Smad3, making the cells sensitive to lipid oxidation [95].

Ferroptosis has also been investigated within inflammatory cells themselves. The iron and ferroptosis may explain differences in macrophage characteristics between the inflammatory M1 and alternative M2 polar forms and their susceptibility to ferroptosis [96,97]. Iron is not only a key regulator of ferroptosis, but it also polarizes macrophages toward the more inflammatory M1 subtype via acetylation of p53 [98]. Experimentally-induced inflammatory macrophages (M1) have been reported to be resistant to ferroptosis due to reduced 15-LOX activity [99]. It is known that mutations in p53 reduce the sensitivity of macrophages to ferroptosis and alter their response to certain types of infection [100].

CD8^+^ T cells have also been reported to suppress the expression of system Xc^−^ in tumor cells via IFNγ secretion, thus promoting ferroptosis [101]. The inhibition of GPX4 induces ferroptosis in CD8^+^ T cells in vitro [102,103]. T cell survival and proliferation are also suggested to be regulated by ferroptosis, as a GPX4-deficient T cell model showed that GPX4 was required for CD4^+^ T cell and CD8^+^ T cell proliferation during infection [103].

The infiltration of various inflammatory cells into cardiac tissue has also been observed during myocardial infarction [104]. Various reports have shown the involvement of T cells and macrophages during the process of post-I/R myocardial remodeling that leads to reduced contractility and left ventricular dilatation [105,106,107]. Taken together, these studies suggest that regulating ferroptosis in both infarcted tissue and inflammatory cells responding to the infarcted site may possibly lead to better overall control of tissue inflammation, since ferroptosis and inflammation can activate each other in turn through cell-mediated inflammation.

## 4. Conclusions

Ferroptosis is a form of regulated cell death caused by the disruption of the balance between lipid peroxidation and antioxidant systems. Rapid progress has been made in discovering mechanisms that promote and inhibit ferroptosis, as well as downstream targets and effects of ferroptosis. In addition, there is an increasing number of reports that suggest that ferroptosis is closely related to I/R injury, especially with regard to cardiac I/R injury. Ferroptosis may be a contributing factor toward inflammation observed in LV remodeling, which in turn determines a patient’s prognosis after cardiac I/R injury. There is also the possibility that cell death caused by cardiac I/R injury is not limited to ferroptosis and that multiple cell death pathways may be simultaneously at work, thus possibly complicating the studies of the relationship between cardiac I/R injury and inflammation. While the relative contribution of ferroptosis towards cell death in I/R injury is yet to be determined, in any case, it is important to minimize the initial extent of cell death that occurs regardless of the types of cell death involved, as cell death is the trigger that initiates the maladaptive mechanisms that occur in response to I/R injury. Therefore, we expect further development and clinical application of therapies that target and suppress ferroptosis at the initial infarct site, as well as inflammation caused by the infarcted tissue and cells recruited to the infarct site.

## Figures and Tables

**Figure 1 cells-11-01399-f001:**
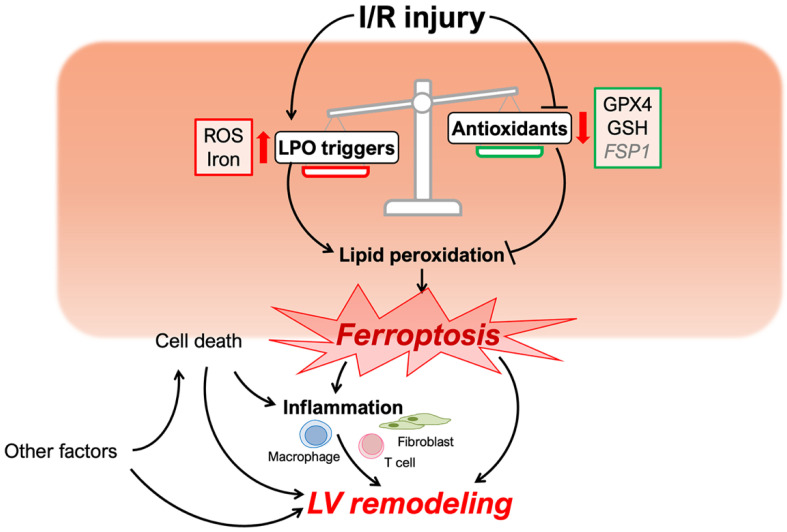
Schematic of ferroptosis and left ventricular remodeling caused by cardiac I/R injury. Cardiac I/R injury triggers the process of lipid peroxidation (LPO) and attenuates inhibitory mechanisms preventing LPO. The disruption of redox homeostasis induces myocyte ferroptosis, which, with other factors, may influence left ventricular remodeling. ROS, reactive oxygen species; LOXs, lipoxygenases; GPX4, glutathione peroxidase 4; GSH, glutathione; FSP1, ferroptosis suppressor protein 1.

**Figure 2 cells-11-01399-f002:**
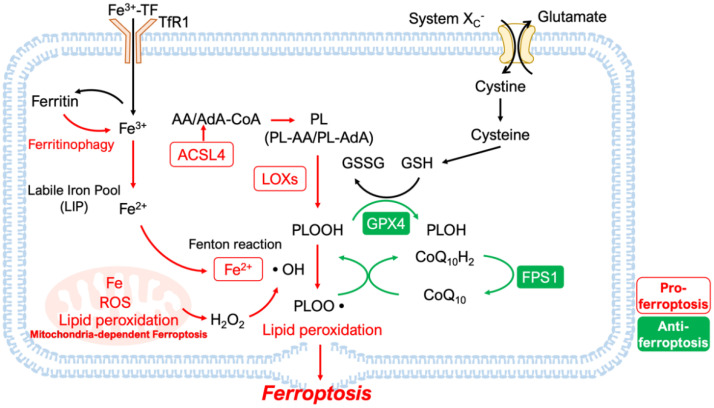
Schematic of the core mechanisms of ferroptosis. Ferroptosis is a form of cell death caused by disrupting the balance between lipid peroxidation (LPO) and its inhibitory mechanisms. In addition to Fe-dependent LPO, the enzymatic formation of phospholipid hydroperoxides (PLOOH) by lipoxygenases (LOXs) also contributes to this mechanism. Incorporating polyunsaturated fatty acids (PUFA) into phospholipids also promotes LPO. Several inhibitory mechanisms against ferroptosis exist, including ferroptosis suppressor protein 1 (FPS1), which reduces phospholipid peroxyl radicals (PLOO•) by reducing coenzyme Q_10_H_2_ (CoQ_10_H_2_), and glutathione peroxidase 4 (GPX4), a key enzyme which reduces PLOOH and depends on glutathione (GSH) as a substrate. GSH requires cystine for its synthesis, which is imported by system Xc^−^ transporters. DMT1, Divalent Metal Transporter 1; TF, transferrin; ROS, reactive oxygen species; •OH, hydroxyl radical; •PL, phospholipid radical; ACSL4, acyl-CoA synthetase long-chain family number 4; AA, arachidonic acid; AdA, adrenoyl acid.

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
