# Peer review of "The Role of Ferroptosis in Adverse Left Ventricular Remodeling Following Acute Myocardial Infarction"

_cells, 2022, doi:10.3390/cells11091399_

Round 1

Reviewer 1 Report

This is a very well-written review discussing the features of ferroptosis, including the underlying mechanisms and its contributions to cell death in the heart during stress. The authors are experts of this subject. The citation is fair and includes recent papers. It is very useful for the readers to understand what ferroptosis is and update their knowledge.

Comments:

The authors could have discussed more regarding how to identify the presence of ferroptosis in the heart and the cardiomyocytes therein. 

In response to ischemia/reperfusion (IR), what is the proportion of cardiomyocyte death by ferroptosis among overall death of cardiomyocytes? When does it happen, within 1-2 hours of IR or with a prolonged time course?

The authors could have discussed Kitakata et al (JMCC, 2021) reporting the involvement of ferroptosis in doxorubicin cardiomyopathy.

The authors could highlight unsolved issues and/or knowledge gaps

Author Response

We greatly appreciate the thoughtful evaluation of our manuscript entitled, "The role of ferroptosis in adverse left ventricular remodeling following acute myocardial infarction".  We are so pleased to hear the enthusiasm on it from reviewers.  In response to the Reviewer’s comments, we have modified the manuscript as detailed below.

Reviewer #1: 

  1. “The authors could have discussed more regarding how to identify the presence of ferroptosis in the heart and the cardiomyocytes therein.”

We thank the Reviewer for this helpful suggestion.  We now add a discussion of how to identify the presence of ferroptosis in the heart and the cardiomyocytes in the new section “3.1.4. Identification of ferroptosis in cardiomyocytes”. – Page 9, Line 1073-1198 in the file with Track Changes.

  1. “In response to ischemia/reperfusion (IR), what is the proportion of cardiomyocyte death by ferroptosis among overall death of cardiomyocytes? When does it happen, within 1-2 hours of IR or with a prolonged time course?”

We have added additional paragraph to discuss the proportion of regulate cell death cardiomyocytes in I/R injury in “3.1.1. Role of iron in I/R injury”. – Page 8, Line 947-958 in the file with Track Changes.

  1. “The authors could have discussed Kitakata et al (JMCC, 2021) reporting the involvement of ferroptosis in doxorubicin cardiomyopathy.”

We appreciate the great comment from the Reviewer.  We have included the Kitakata et al reference (Ref #61) and discussed the topic on Page 6, Line 759-762 in the file with Track Changes.

  1. “The authors could highlight unsolved issues and/or knowledge gaps”

We now discuss unsolved issues and/or knowledge gaps regrading ferroptosis in the heart in each section.  We have added discussions about mitochondria-dependent mechanisms of ferroptosis on Page 6, Line 775-778, and the proportion of cardiomyocyte death by ferroptosis in I/R injury on Page 8, Line 955-958.  On that note, we agree that there are still knowledge gaps in ferroptosis in the heart and highlighted this further in our manuscript that there is still much more yet to be discovered with ferroptosis.

Reviewer 2 Report

The MS by Komai et al summarizes and discusses previous studies on the role of ferroptosis in the pathogenesis of post-MI cardiac remodeling. The MS tries to explain the main pathways involved in ferroptotic cell death during MI. The MS has several weaknesses and requires extensive revision to improve its readability, presentation, and significance.

Major comments:

  1. 1. Introduction lacks a logistic order in terms of history, definition and current status of ferroptosis followed by its role in post-MI. First paragraph (lines 24-45): Despite “ferroptosis” being first invented in 2012, several characteristics/ components of ferroptotic cell death were demonstrated earlier (PMID: 2576375; PMID: 11895126). A brief analysis of previous studies that predicted the existence of a new cell death mechanism named later as “ferroptosis” should be discussed. Also, here the main characteristics and components (GSH, GPX4, LOX, ACSL4) should be given briefly.
  2. 3.1. Cardiomyocyte cell death by ferroptosis (line 190-293). Pioneer studies indicating ferroptosis in the heart should be discussed. Currently, all known anti-ferroptotic compounds (such as Fer-1, Lip-1, XJB-5-131, MitoTEMPO) are not specific inhibitors of ferroptosis; they are antioxidants, and cardioprotective effects of these compounds do not provide direct evidence on their anti-ferroptotic effects and can be explained by their cellular/mitochondrial antioxidant action. Likewise, reduction of GPX4 activity, increased iron levels and lipid peroxidation do not indicate a direct involvement of ferroptosis. Only a few studies that analyzed and quantified ferroptotic oxidized phosphatidylethanolamine (oxPE) by detailed lipidomics provided direct evidence of ferroptosis in cardiomyocytes in myocardial infarction (PMID: 30830879) and ischemia-reperfusion (PMID: 34102574). Furthermore, oxPE were visualized in cardiomyocytes through direct mapping of phospholipid ferroptotic death signals by gas cluster ion beam secondary ion mass spectrometry (PMID: 33684237). Indeed, the last was first to demonstrate ferroptotic oxygenated PL species in cardiac cells.
  3. Lines 108-110: “Divalent Metal Transporter 1 (DMT1) is an additional transmembrane protein for the uptake of ferrous iron (Fe2+) into cells” is not correct since DMT1 is responsible for the release of Fe2+ from the endosome to the cytoplasm (PMID: 33925597).
  4. 2.2.3. Mitochondria-dependent ferroptosis (lines 176-188). The role of mitochondria in ferroptosis has been discussed recently in several review papers. The authors could highlight the role of mitochondria in ferroptosis in response to MI/cardiac IR, and describe briefly the crosstalk between mito ROS, redox status (GSH/GSSG), mito iron metabolism, and ferroptosis in the heart. Conclude this section with “The role of mitochondria in ferroptosis has been reviewed and discussed in detail elsewhere (PMID: 31636403; PMID: 33801920; PMID: 35342847; PMID: 34328510)”
  5. 3.2. Ferroptosis-mediated inflammation in I/R injury (lines 294-375). There are not many studies on the role of inflammation in ferroptotic signaling in post-MI hearts. These section should be shortened and mainly focused on the studies that elucidated the role of inflammatory factors in ferroptotic signaling pathways.

Minor comments:

  1. Line 135: Change “…Cysteine is an essential part of the molecular structure of GSH” to “Cysteine along with glycine and glutamate participate in GSH synthesis”.
  2. The MS needs to be revised to correct sentence style, and avoid repeats (e.g., lines 76-77 and 78-79).

Author Response

We greatly appreciate the thoughtful evaluation of our manuscript entitled, "The role of ferroptosis in adverse left ventricular remodeling following acute myocardial infarction".  We are so pleased to hear the enthusiasm on it from reviewers.  In response to the Reviewer’s comments, we have modified the manuscript as detailed below.

Reviewer #2:

Major comments:

  1. Introduction lacks a logistic order in terms of history, definition and current status of ferroptosis followed by its role in post-MI. First paragraph (lines 24-45): Despite “ferroptosis” being first invented in 2012, several characteristics/ components of ferroptotic cell death were demonstrated earlier (PMID: 2576375; PMID: 11895126). A brief analysis of previous studies that predicted the existence of a new cell death mechanism named later as “ferroptosis” should be discussed. Also, here the main characteristics and components (GSH, GPX4, LOX, ACSL4) should be given briefly.”

            We thank the Reviewer for their insightful and constructive comments.  We have retooled the introduction, Page 1-3, to include a more chronological narrative per the Reviewer’s recommendations.  We have cited Ref #7 for PMID: 2576375 and Ref #8 for PMID: 11895126 and discussed oxidative stress regulated by GSH in the introduction.

  1. “3.1. Cardiomyocyte cell death by ferroptosis (line 190-293). Pioneer studies indicating ferroptosis in the heart should be discussed. Currently, all known anti-ferroptotic compounds (such as Fer-1, Lip-1, XJB-5-131, MitoTEMPO) are not specific inhibitors of ferroptosis; they are antioxidants, and cardioprotective effects of these compounds do not provide direct evidence on their anti-ferroptotic effects and can be explained by their cellular/mitochondrial antioxidant action. Likewise, reduction of GPX4 activity, increased iron levels and lipid peroxidation do not indicate a direct involvement of ferroptosis. Only a few studies that analyzed and quantified ferroptotic oxidized phosphatidylethanolamine (oxPE) by detailed lipidomics provided direct evidence of ferroptosis in cardiomyocytes in myocardial infarction (PMID: 30830879) and ischemia-reperfusion (PMID: 34102574). Furthermore, oxPE were visualized in cardiomyocytes through direct mapping of phospholipid ferroptotic death signals by gas cluster ion beam secondary ion mass spectrometry (PMID: 33684237). Indeed, the last was first to demonstrate ferroptotic oxygenated PL species in cardiac cells.”

We also thank the reviewer for pointing out corrections and details about the mechanisms underlying ferroptosis and anti-ferroptotic compounds, and agree that there are few, but very significant and important studies focusing on ferroptosis on the heart.  We have integrated the suggested references into the manuscript to further clarify and improve the accuracy of the manuscript.  We have cited Ref #22 for PMID: 30830879 and added additional discussion about OxPE in heart grafts in a new paragraph, Page 9, Line 1057-1064.  We have also cited Ref # 59 for PMID: 34102574 and added a discussion of the contribution of mitochondrial glutathione to ferroptosis in cardiomyocytes.  We have added a new discussion of an advanced technique, gas cluster ion beam secondary ion mass spectrometry (GCIB-SIMS) imaging, for analyzing OxPEs in myocardial cell lines with Ref #84 (PMID: 33684237).

  1. “Lines 108-110: “Divalent Metal Transporter 1 (DMT1) is an additional transmembrane protein for the uptake of ferrous iron (Fe2+) into cells” is not correct since DMT1 is responsible for the release of Fe2+ from the endosome to the cytoplasm (PMID: 33925597).

            We agree with the Reviewer.  Recent reports suggest DMT1 is also responsible for the release of Fe2+ from the endosome to the cytoplasm.  We have cited Ref #42 (PMID: 33925597) and Ref # 41 to discuss the mechanisms of DMT1 on iron transport on Page 5, Line 418-420.  We have also modified Figure 2 accordingly.

  1. “2.2.3. Mitochondria-dependent ferroptosis (lines 176-188). The role of mitochondria in ferroptosis has been discussed recently in several review papers. The authors could highlight the role of mitochondria in ferroptosis in response to MI/cardiac IR, and describe briefly the crosstalk between mito ROS, redox status (GSH/GSSG), mito iron metabolism, and ferroptosis in the heart. Conclude this section with “The role of mitochondria in ferroptosis has been reviewed and discussed in detail elsewhere (PMID: 31636403; PMID: 33801920; PMID: 35342847; PMID: 34328510)”

            We thank the Reviewer for this helpful suggestion.  We have added more about mitochondria-depended ferroptosis in the section 2.2.3, Page 6, to discuss mitochondria-dependent ferroptosis in detail, using the recommended references, Ref #54 (PMID: 31636403), Ref #56 (PMID: 35342847), Ref #57 (PMID: 34328510) and additional recent publications, Ref #59 (Jan, S. et al. Redox Biol 2021) and Ref #62 (Mao, C. et al. Nature 2021).

  1. “3.2. Ferroptosis-mediated inflammation in I/R injury (lines 294-375). There are not many studies on the role of inflammation in ferroptotic signaling in post-MI hearts. These section should be shortened and mainly focused on the studies that elucidated the role of inflammatory factors in ferroptotic signaling pathways.”

We thank the Reviewer for their insightful and constructive comments.  We have reduced the emphasis on the section regarding the role of inflammation in ferroptotic signaling in post-MI hearts in 3.2.  We have changed the subtitle to “Ferroptosis and inflammation during I/R injury” and focused on the role of inflammatory factors in ferroptotic signaling pathways.

Minor comments:

  1. “Line 135: Change “…Cysteine is an essential part of the molecular structure of GSH” to “Cysteine along with glycine and glutamate participate in GSH synthesis”.

We thank the Reviewer for the comments.  We changed it accordingly.  Line 445 in the file with Track Changes.

  1. “The MS needs to be revised to correct sentence style, and avoid repeats (e.g., lines 76-77 and 78-79).

We thank the Reviewer for the comments.  We corrected that.  We have proofread the manuscript and edited it to make the narrative less repetitive.

Round 2

Reviewer 2 Report

The authors significantly revised the MS and addressed most of my comments.

Minor:  For all abbreviations, full names should be given on the first mention in the text followed by the abbreviation in the bracket (e.g., OxPCs - line 390).